# Neutrophil Elastase, Neuron-Specific Enolase, and S100B Protein as Potential Markers of Long-Term Complications Caused by COVID-19 in Patients with Type 2 Diabetes Mellitus (T2DM) and Advanced Stage of Diabetic Nephropathy (NfT2DM)—Observational Studies

**DOI:** 10.3390/ijms252111791

**Published:** 2024-11-02

**Authors:** Maciej Rabczyński, Sandra Chwałek, Joanna Adamiec-Mroczek, Łukasz Lewandowski, Małgorzata Trocha, Beata Nowak, Marta Misiuk-Hojło, Dorota Bednarska-Chabowska, Edwin Kuźnik, Paweł Lubieniecki, Joanna Kluz, Zofia Kaszubowska, Mikołaj Kondracki, Wojciech Grodzki, Jakub Federowicz, Magdalena Mierzchała-Pasierb, Andrzej Gamian, Agnieszka Bronowicka-Szydełko, Katarzyna Madziarska

**Affiliations:** 1Clinical Department of Diabetology, Hypertension and Internal Diseases, Wroclaw Medical University, Borowska Street 213, 50-556 Wroclaw, Poland; maciej.rabczynski@umw.edu.pl (M.R.); malgorzata.trocha@umw.edu.pl (M.T.); dorota.bednarska-chabowska@umw.edu.pl (D.B.-C.); edwin.kuznik@umw.edu.pl (E.K.); pawel.lubieniecki@umw.edu.pl (P.L.); joanna.kluz@umw.edu.pl (J.K.); katarzyna.madziarska@umw.edu.pl (K.M.); 2Faculty of Chemistry, Wroclaw University of Science and Technology, 27 Wybrzeże Stanisława Wyspiańskiego St., 50-370 Wroclaw, Poland; 276114@student.pwr.wroc.pl; 3Clinical Department of Ophthalmology, Wroclaw Medical University, Borowska Street 213, 50-556 Wroclaw, Poland; joanna.adamiec-mroczek@umw.edu.pl (J.A.-M.); marta.misiuk-hojlo@umw.edu.pl (M.M.-H.); 4Department of Biochemistry and Immunochemistry, Wroclaw Medical University, Chałubińskiego Street 10, 50-368 Wroclaw, Poland; lukasz.lewandowski@umw.edu.pl (Ł.L.); magdalena.mierzchala-pasierb@umw.edu.pl (M.M.-P.); 5Department of Pharmacology, Wroclaw Medical University, ul. Jana Mikulicza-Radeckiego 2, 50-345 Wrocław, Poland; beata.nowak@umw.edu.pl; 6Faculty of Medicine, Wroclaw Medical University, Pastura 1, 50-368 Wroclaw, Poland; zofia.kaszubowska@student.umw.edu.pl (Z.K.); mikolaj.kondracki@student.umw.edu.pl (M.K.); wojciech.grodzki@student.umw.edu.pl (W.G.); jakub.federowicz@student.umw.edu.pl (J.F.); 7Laboratory of Medical Microbiology, Hirszfeld Institute of Immunology and Experimental Therapy, Polish Academy of Sciences, 53-114 Wroclaw, Poland; andrzej.gamian@hirszfeld.pl

**Keywords:** advanced diabetic nephropathy, COVID-19, neutrophil elastase, neuron-specific enolase, post-COVID-19, S100B, type 2 diabetes

## Abstract

Despite numerous studies conducted by various research teams, predicting long-term outcomes (known as Post-COVID-19 Syndrome, PCS) that may result from Coronavirus Disease 2019 (COVID-19) remains challenging. PCS affects over a million people, primarily those with comorbid conditions. Therefore, it is crucial to undertake research aimed at developing a predictive model for early diagnosis of PCS, which in turn would enable faster preventive actions. The aim of this study was to assess the value of measuring and attempt a quantitative evaluation using Enzyme-Linked Immunosorbent Assay (ELISA) tests of three non-serum proteins, whose presence in the blood during COVID-19 was associated with severe disease progression: neutrophil elastase (NE), calcium-binding protein S100B, and neuron-specific enolase (NSE). The concentrations of these proteins were measured in blood serum samples collected before the COVID-19 pandemic from (1) patients with type 2 diabetes (T2DM); (2) advanced stage diabetic nephropathy (NfT2DM); (3) a healthy group; and in blood serum samples collected two years after recovering from COVID-19 from patients with (4) T2DM and (5) NfT2DM. It was found that elevated levels of NE and NSE were significantly more common (*p* < 0.05) in patients with NfT2DM after recovering from COVID-19 compared to the other groups, while elevated levels of S100B were significantly more frequently observed in patients with T2DM after recovering from COVID-19 (*p* < 0.05). Demonstrating differences in the prevalence of NE, NSE, and S100B in individuals who recovered from COVID-19 with T2DM and NfT2DM makes these proteins important components of the developing predictive model for early detection of PCS. To our knowledge, this is the first study showing the significance of NE, NSE, and S100B in PCS in the context of T2DM and NfT2DM.

## 1. Introduction

Neutrophil elastase (NE), calcium-binding protein S100B, and neuron-specific enolase (NSE) are proteins with potential diagnostic significance. Although these proteins are not normally present in human blood or are present only in trace amounts, they can be released into the blood at high concentrations in certain specific metabolic disorders. This makes them fairly specific markers in diagnosing these disorders.

NE is a serine protease found in neutrophils and extracellular neutrophil traps (NETs), released as a result of neutrophil activation, such as in response to trauma, inflammation, or infection [1] (Figure 1a). Excessive activity of this enzyme disrupts immune system function, intensifies inflammatory processes, causes multi-organ damage, and promotes thrombosis—NE has been shown to be a determinant of thrombosis and a severe course of COVID-19 [2]. Elevated neutrophil counts and the presence of NETs in the blood are predictive factors for a more severe course of COVID-19 [3], while NE is considered a potential biomarker for multi-organ changes caused by COVID-19 due to the so-called “cytokine storm” and coagulopathy [4]. The cytokine IL-1ß secreted by macrophages stimulates neutrophils to form NETs [3]. During COVID-19, the level of interleukin 8 (IL-8)—a chemotactic factor that recruits neutrophils and directs them to infected sites—also increases, which promotes NET synthesis [5]. NETs are areas where erythrocytes, platelets, and fibrin—elements necessary for clot formation—accumulate. Tissue factor (TF), present in these NETs, induces the coagulation cascade via the extrinsic pathway [6]. Furthermore, NE degrades collagen and elastin fibers, damaging the extracellular matrix of the lungs, intensifying mucus secretion, and disrupting ciliary movement, which also contributes to exacerbating COVID-19 symptoms [1].

The S100B protein is expressed in various types of cells, including protoplasmic astrocytes and myelinating oligodendrocytes [7]. Elevated levels of this protein, such as in cerebrospinal fluid, blood, urine, saliva, and amniotic fluid, can indicate brain damage, neuroinflammatory or psychiatric disorders [8]. It has been shown that S100B correlates with the extent of trauma, survival, and neurological parameters [9], and increased expression of this protein in the blood serum of patients with acute exacerbations and deficit symptoms in schizophrenia [10] may lead to a progressive reduction in neuropil [11]. Overexpression of S100B is observed in various conditions, including Down syndrome [12], Alzheimer’s disease [13], epilepsy [14], Parkinson’s disease [15], multiple sclerosis [16], schizophrenia [17], cerebral palsy [18], Creutzfeldt–Jakob disease [19], autism spectrum disorder [20], and also in COVID-19 [21]. The molecular consequences of S100B action are shown in Figure 1b.It has been shown that one of the main proteins of the SARS-CoV-2 virus (the so-called spike protein) stimulates the secretion of S100B through the activation of receptors on microglial cells [21]. Additionally, a significant correlation has been demonstrated between the level of SARS-CoV-2 and the clinical severity of the disease [22]. However, it should be noted that the S100 family has a negative impact on brain health [23] through the formation of neurotoxic fibrils [24].

NSE is primarily found in neurons and activates various cellular pathways responsible for proliferation, growth, and differentiation of cells (mitogen extracellular kinase/extracellular regulated protein kinase, MAPK/ERK) [25], maintaining cell viability (phosphoinositide 3-kinases/protein kinase B, PI3K/Akt) [26], stabilizing the cytoskeleton by regulating actin polymerization (Ras homolog family member A/Rho-associated protein kinase, RhoA/ROCK) [27], as well as in signaling pathways that enhance inflammatory states (Figure 1c). Following injury, NSE migrates to the cell surface and stimulates the production of reactive oxygen species (ROS), nitric oxide (NO), and various pro-inflammatory cytokines, such as TNF-α, IL-1β, INF-γ, TGF-β, and MCP-1 [28]. NSE is a diagnostic protein for neurological disorders (tuberculous meningitis [29], Alzheimer’s disease, Parkinson’s disease, amyotrophic lateral sclerosis [30], Creutzfeldt–Jakob disease, Guillain–Barré syndrome, hypoxic–ischemic encephalopathy, intracranial injuries [31], and epilepsy [32]), neuroendocrine tumors, and neuroblastoma [29], as well as various respiratory diseases, such as lung cancer, tuberculosis, acute lung injury, and chronic obstructive pulmonary disease [28]. NSE levels are one of the parameters of stroke severity—rapid declines in NSE concentration are characteristic of mild changes, while increased NSE levels are associated with poor prognosis [33]. NSE is also an important marker for COVID-19. It has been shown that children with neurological symptoms in COVID-19 (e.g., headaches, consciousness disturbances, and seizures) had significantly higher levels of NSE compared to the group without complications [34], and elevated NSE levels were observed in patients with severe pulmonary failure, often requiring respiratory support [35].

PCS (Post-COVID-19 Syndrome) refers to symptoms persisting for more than 3 months after recovering from COVID-19, leading to prolonged changes in individual organs or multi-organ changes [34]. These symptoms include difficulty concentrating, cognitive dysfunction, amnesia, depression, fatigue, and anxiety [36], with risk factors for persistent neuropsychiatric symptoms in PCS including older age, female sex, and the severity of comorbid conditions such as diabetes [37], which is often associated with tachycardia, sarcopenia, microcirculation dysfunction, or organ damage [38,39]. This necessitates the development of optimal predictive models to identify patients in the highest risk groups for acute and long-term complications in order to take preventive actions, thereby reducing the risk of death and long-term complications.

The aim of the study was to determine whether there is any justification for measuring NE, S100B, and NSE in patients with T2DM and NfT2DM who have recovered from COVID-19, suggesting that these proteins might play a role in building a predictive model for PCS. Thus, an attempt was made to measure NE, S100B, and NSE levels using ELISA tests in blood serum samples collected before the COVID-19 pandemic from patients with (1) T2DM and (2) advanced-stage diabetic nephropathy (NfT2DM); (3) a healthy group; and in blood serum samples collected two years after recovering from COVID-19 from patients with (4) T2DM and (5) NfT2DM. It was also necessary to determine the reference values for NE and S100B, considering that these proteins are not physiologically present in blood (or are present in trace amounts)—to our knowledge, these values for NE and S100B have not yet been determined. The studies conducted were preliminary, guiding further research to be carried out on a significantly larger patient group, and the obtained results should be compared with values from standard laboratory diagnostics, imaging, clinical diagnoses, and applied therapies.

## 2. Results

### 2.1. Quantitative Assessment of Neutrophil Elastase (NE) in Blood Serum

To enable the determination of concentrations of the analytes being studied, such as NE, NSE, or S100B, it was necessary to establish standard curve profiles. Since commercial ELISA kits were used for the tests, the concentrations of the standards (i.e., NE, NSE, or S100B) were prepared according to the manufacturer’s instructions, and the curves were determined as a relationship between absorbance measured at λ = 450 nm and concentration. Example values of NE concentrations are shown in Figure 2.

Based on the obtained results, it can be stated that in 13.5% of patients, NE concentration was below the test’s sensitivity threshold, i.e., <31 pg/mL. Since NE is an indicator enzyme rather than a secretory one (meaning it is released into the blood only upon cell damage, such as NE from neutrophils), a cutoff point, which is the average value of the reference range, was set at 385 pg/mL—somewhat lower than half of the value established by the MyAssays program, a tool used for analyzing data in biological tests. This program set 780 pg/mL as the cutoff, separating the dataset into two groups (with the lowest and highest results). However, it was clear that the threshold value should be significantly lower due to the indicator nature of the enzyme. This was also because the studies were preliminary and pilot in nature, checking whether any changes in NE concentrations could be observed in patient groups before and after COVID-19. Thus, a value of 385 pg/mL, which is half the previously suggested threshold, was proposed as a tentative value, the validity of which needs to be confirmed in studies conducted on a much larger patient group.

NE values above 385 pg/mL were observed in 54.23% of patients with T2DM before COVID-19. Additionally, the analysis revealed that the average NE concentration was 455 pg/mL, with a maximum value of 1105 pg/mL. The high standard deviation (SD) indicates that the arithmetic mean is not a reliable result for determining the reference range. Furthermore, in 13 out of 59 patients, NE concentrations were above 780 pg/mL, representing 22% of all patients with T2DM before COVID-19. Based on the conducted studies, it was found that 18 out of 62 (29%) patients with type 2 diabetes, after recovering from COVID-19, had NE concentrations outside the test’s sensitivity range (c < 31 pg/mL), while 17 patients had levels above 780 pg/mL (27%). The results showed that the number of patients with NE > 780 pg/mL and type 2 diabetes before COVID-19 was 5 percentage points (p.p.) lower compared to the group of T2DM patients after COVID-19. An increase in the number of patients was observed for NE concentrations > 385 pg/mL—36 out of 62 patients with T2DM after COVID-19 (56%) had NE levels exceeding 385 pg/mL, which also indicates a slight increase in the number of patients with NE > 385 pg/mL (~2 p.p.) compared to the group with T2DM before COVID-19. Moreover, in patients with T2DM without NfT2DM who recovered from COVID-19, the average NE concentration was 481.02 pg/mL, representing an increase of 122 pg/mL compared to the pre-COVID-19 group.

The highest NE value in the group of patients with NfT2DM before COVID-19 was 1381 pg/mL. Based on the conducted studies, it was found that 16 out of 55 (29.1%) patients with NfT2DM had NE concentrations above 780 pg/mL, while 8 out of 51 (16%) had values below the test’s sensitivity. NE > 385 pg/mL was present in 30 out of 51 patients (58.8%). The average NE concentration was 585.7 pg/mL (SD = 691 pg/mL, with the high SD primarily due to the very high maximum value of 4390 pg/mL), which was 130 pg/mL higher than the average NE concentration in the pre-COVID-19 T2DM patient group without NfT2DM. Excluding the highest NE value of 4390 pg/mL, which significantly differed from the other results, reduced the average NE concentration to 509.6 pg/mL, which is 54 pg/mL higher than the average NE concentration in the pre-COVID-19 T2DM patient group without NfT2DM. Based on the studies, it was found that 3 out of 55 patients with NfT2DM who recovered from COVID-19 had NE concentrations below the test’s sensitivity (c < 31 pg/mL), while 6 out of 55 (9%) patients with NfT2DM had NE concentrations above 780 pg/mL. The average NE concentration was 467.0 pg/mL, lower than in the corresponding pre-COVID-19 group. Interestingly, NE > 385 pg/mL was observed in 39 out of 55 patients (70.9%), indicating a significant increase (12.1 p.p.) in the number of patients with markedly elevated NE concentrations, which could have been caused by recovering from COVID-19, especially since patients from both groups with NfT2DM were of similar age.

Based on the conducted studies, it was noted that 3 out of 89 (3.09%) healthy individuals from whom blood was collected before COVID-19 had NE concentrations below the test’s sensitivity (c < 31 pg/mL), while 13 (13.4%) had NE concentrations above 780 pg/mL. The average NE concentration was 462.0 pg/mL, comparable to the average NE concentration in the pre-COVID-19 T2DM patient group. NE > 385 pg/mL was found in 50 out of 89 individuals (56.2%). From the studies, it can be observed that the percentage of patients with NE > 385 pg/mL was similar across all patient groups, i.e., in the pre-COVID-19 T2DM group (54.2%) and post-COVID-19 T2DM group (56.45%), in the NfT2DM group before COVID-19 (58.8%), and in the group of healthy blood donors. However, there was a significantly higher percentage of patients with NE > 385 pg/mL (70.2%) in the nephropathy group after COVID-19. This observation is illustrated in Figure 3.

Based on the presented graph, it can be inferred that for the threshold value of c = 385 pg/mL, the percentage of individuals with NfT2DM is significantly higher in the post-COVID-19 group compared to the pre-COVID-19 group (*p* < 0.05). This result suggests a potential impact of COVID-19 on the increase in the number of individuals with notably elevated NE levels (c > 385 pg/mL). Such an observation was not noted for the T2DM patients without NfT2DM before and after COVID-19. This observation indicates a possible influence of impaired blood vessel condition, due to microangiopathy, such as nephropathy, on the increase in NE levels.

### 2.2. Quantitative Assessment of S100B Protein in Blood Serum

To enable the determination of S100B levels, it was necessary, similar to NE, to establish standard curve profiles. These curves were prepared according to the manufacturer’s instructions as a function of absorbance values measured at λ = 450 nm against concentration.

The results of the study showed that the S100B concentration in serum from patients with T2DM, from whom blood was collected before COVID-19, was detectable (≥7.8 pg/mL) in 7 out of 59 patients with T2DM, representing 11.9% of the entire group. The study results indicated that S100B concentration was measurable (c ≥ 7.8 pg/mL) in 49 out of 62 patients with T2DM who had recovered from COVID-19, representing a significant 79% of the group. In the remaining samples, S100B levels were below the test sensitivity, i.e., 7.8 pg/mL. Additionally, the study results showed that S100B concentration was detected in the serum of only 1 out of 51 patients with NfT2DM before the COVID-19 pandemic, accounting for less than 2% of the studied group. For the remaining patients, S100B levels were below the test sensitivity, i.e., 7.8 pg/mL. Based on the obtained results, it can be stated that S100B concentration was detected in the serum of 6 out of 55 (10.9%) patients with NfT2DM who had recovered from COVID-19. This indicates an increase in the number of patients with NfT2DM in whom S100B was released into the serum. Furthermore, S100B concentration was detected in the serum of 2 out of 91 (2.20%) individuals in the control group, from whom blood was collected before the COVID-19 pandemic (Figure 4).

Based on the presented chart, it can be inferred that for the threshold value of S100B ≥ 7.8 pg/mL, the percentage of individuals with T2DM is significantly higher in the post-COVID-19 group compared to the pre-COVID-19 group (*p* < 0.05). This result suggests a potential impact of COVID-19 on the increase in the number of individuals with markedly elevated levels of S100B (c > 7.8 pg/mL). Such a significant increase was not observed in the groups of patients with NfT2DM before and after COVID-19.

### 2.3. Quantitative Assessment of NSE Protein in Blood Serum

Concentrations of neuron-specific enolase (NSE) in the tested samples were determined using a diagnostic test based on standard curves prepared according to the manufacturer’s guidelines. These curves were constructed as a function of changes in absorbance measured at λ = 450 nm relative to concentration. An elevated NSE level outside the reference range was defined in the test as a value of c ≥ 12 ng/mL.

The average NSE concentration in the serum of patients with NfT2DM before the COVID-19 pandemic was 7.4 ng/mL, similar to that of the T2DM group before COVID-19. Elevated NSE levels were found in 6 out of 50 (12%) patients, with 88% having levels within the reference range. The results showed that NSE levels were above the reference values in 25 out of 54 serum samples from patients with NfT2DM who had recovered from COVID-19, representing 46.3% of the group. The proportion of patients with elevated NSE was therefore approximately 34 percentage points higher than in the group with NfT2DM from before the COVID-19 pandemic. The average NSE concentration in this group was significantly higher at 12.1 ng/mL, which exceeded the reference values for NSE. The average NSE concentration in patients with NfT2DM after recovering from COVID-19 was significantly higher than in the group with NfT2DM before COVID-19, showing a difference of 4.6 ng/mL, indicating a 62% increase compared to the average NSE concentration in the pre-COVID-19 NfT2DM group. Furthermore, the results showed that NSE levels were above the reference values in 15 out of 90 serum samples from the control group, representing 16.7% of the group. A comparison of the percentage of patients with NSE concentrations ≥12 ng/mL in each group is presented in Figure 5.

Based on the presented chart, it can be inferred that for NSE ≥ 12 ng/mL, the percentage of patients with NfT2DM is significantly higher in the post-COVID-19 group compared to the pre-COVID-19 group (*p* < 0.05). This result suggests a potential impact of COVID-19 on the increase in the number of individuals with distinctly elevated NSE levels (c ≥ 12 ng/mL). In comparison, there was a slight decrease in the number of patients with elevated NSE levels beyond the reference range in the T2DM groups before and after COVID-19, with a difference of 10.5 percentage points. Therefore, average concentrations were compared across all patient groups, as shown in Figure 6.

The results of the comparison of average NSE concentrations in all groups showed that these values were comparable, except for the group of patients with NfT2DM after recovering from COVID-19. In this group, the average NSE concentration was not only significantly higher than the average NSE concentrations in the other groups, but also exceeded the reference values. In the groups of patients with T2DM both before and after COVID-19, the average NSE concentrations were within the reference range. Similarly, the average NSE concentrations in the NfT2DM group and the control group were within the reference range.

Additionally, an analysis was conducted comparing the percentage of patients in each group who showed an increase above the accepted threshold values (i.e., NE > 385 pg/mL, S100B ≥ 7.8 pg/mL, and NSE ≥ 12 µg/mL) for at least two of the measured proteins. The results are presented in Figure 7.

Based on the analysis, it was found that in the group of patients with NfT2DM who had recovered from COVID-19, 5.5% exhibited elevated levels of all three measured proteins (NE, S100B, and NSE) above the established reference values. No other group had any patients with simultaneous elevations above the established threshold values for NE, S100B, and NSE. Furthermore, in this group, 30.9% of patients showed simultaneous increases beyond the reference ranges for NE and NSE (compared to less than 10% in other groups). These differences were statistically significant (*p* < 0.05). In the group of patients with T2DM who had recovered from COVID-19, there was also a significantly higher occurrence of elevated levels above the established reference values for NE and S100B.

### 2.4. Between-Sex Differences in Concentration of NE, NSE, S100B, Age, and Creatinine: Significant Correlations Analyzed Between NE and NSE with Age, Total Protein, Urea, and Creatinine Among the Studied Groups

The baseline values of key anthropometric and biochemical parameters for patient groups 1 to 5 are presented accordingly in Appendix A. There were no differences between men and women in regard to NE, NSE, and S100B (Table 1). However, women among the two NfT2DMs showed lower age compared to men (*p* = 0.038 and *p* = 0.013, respectively). Moreover, the women in the groups T2DM after COVID-19 and the control group showed higher creatinine concentration compared to the men (*p* = 0.006 and *p* = 0.002, respectively).

A significant (*p* < 0.05) weak, positive Spearman correlation was found in NfT2DM patients, before COVID-19, between creatinine and NE (ρ = 0.314). Moreover, the T2DM group (before COVID-19) showed weak negative correlations between creatinine and both NE and NSE (ρ = −0.296 and ρ = −0.274, respectively), as shown in Table 2.

## 3. Discussion

NE, S100B, and NSE are proteins whose potential significance has been demonstrated in patients with COVID-19. Given the existing risk of post-COVID-19 sequelae (PSC), especially in individuals with comorbidities such as type 2 diabetes (T2DM), an attempt was made to measure these proteins in the blood serum of patients with T2DM, including those with nephrologically complicated T2DM (i.e., NfT2DM IV or V, i.e., eGFR < 30), both before COVID-19 and in patients who had recovered from COVID-19 (documented positive test for SARS-CoV-2 mRNA). ELISA tests were also performed on blood serum samples from healthy individuals collected just before the COVID-19 pandemic (these were volunteer blood donors from the Regional Blood Donation Center). All blood groups had similar sample sizes (about 60 samples), except for the samples from the Regional Blood Donation Center, which numbered 98, and all groups consisted of individuals of similar age. The ratio of women to men in the groups was approximately 1:1. It was known that the levels of NSE and S100B in blood, i.e., “brain proteins”, are not gender-dependent but age-dependent. The average increase in the concentration of these proteins is about 1% per year, with high levels of NSE or S100B appearing more frequently in older individuals with dementia [40]. Therefore, it is presumed that high concentrations of these proteins are rare in healthy individuals and indicate neurological disorders. Participants in each group of the study were primarily middle-aged and older adults (mean age: 55 years). It was shown that in the group of healthy individuals, CS100B ≥ 7.8 pg/mL was detectable in only 2% of participants, which may result from emerging neurological disorders, especially in one of two individuals with very high S100B levels (11,270 pg/mL). However, no simultaneous elevation of S100B and NSE was detected in any healthy individual.

In patients with T2DM who had recovered from COVID-19, it was demonstrated that, compared to the T2DM group before COVID-19, the detection rate of S100B increased significantly by 6.6 times (from 11.86% to 79%), and in individuals with NfT2DM after COVID-19 compared to those with NfT2DM before COVID-19—by 5.5 times (from 1.96% to 10.9%). Although the highest S100B concentration in the group of T2DM patients whose blood was collected after COVID-19 was only 700 pg/mL, this protein was detectable at CS100B ≥ 7.8 pg/mL in 79% of participants in the group. Interestingly, it was also observed that simultaneous elevation of S100B and NSE occurred in both of these groups (i.e., with T2DM and NfT2DM after COVID-19), but it was not observed in the groups of individuals with diabetes before COVID-19. The presence of S100B at the level of CS100B ≥ 7.8 pg/mL was surprisingly detected in few patients in the NfT2DM group before COVID-19 (1.96%) and after COVID-19 (10.9%), which might be due not so much to their pharmacotherapy but to the absence of cognitive impairments. This would be consistent with observations from other research teams, which showed that significant increases in S100B levels in patients with end-stage renal disease (compared to a control group) are associated with complications of the disease, such as cognitive impairment [41]. It is also known that S100B levels in patients with end-stage renal disease increase with a decrease in glomerular filtration rate, and S100B levels depend on the timing of blood collection in relation to dialysis [42]. It was noted that the median S100B concentration in serum after dialysis (17.4 µg) is significantly higher than before dialysis (5 µg) in patients with chronic kidney disease [43]. It can thus be concluded that dialysis affects the increase in S100B levels, possibly due to oxidative stress resulting from dialysis, as S100B at high concentrations activates RAGE-dependent overproduction of reactive oxygen species [44]. The literature also presents results showing that S100B levels are lower in patients undergoing hemodialysis and peritoneal dialysis compared to healthy individuals [45]. These observations are consistent with the results presented in this work. In the case of the NfT2DM, blood was collected before dialysis, which may have also contributed to the lower S100B levels. Any attempt to draw conclusions from the results of S100B measurement in the nephropathy patient group should therefore be first analyzed considering the timing of dialysis relative to blood collection.

Moreover, it is known that metabolic disorders characterized by hyperglycemia, dyslipidemia, hypertension, chronic inflammation, or insulin resistance, especially in diabetes (regardless of type) and obesity, often show higher S100B levels than in the control group. It has been shown that there is a correlation between S100B levels and the presence of each of these conditions [46]. This protein is elevated in the blood serum of individuals with abdominal obesity, high triglyceride levels, and decreased insulin sensitivity. Interestingly, the level of this protein increases with weight gain and decreases during prolonged fasting. In metabolic syndrome, S100B levels are higher than in healthy individuals [46]. The results of S100B measurement in this study confirm this observation. Significantly higher S100B levels were observed in the T2DM group both before and after COVID-19 compared to healthy individuals. In the case of patients with NfT2DM, such a clear dependence was observed in those who recovered from COVID-19. It was also shown that in patients with metabolic disorders after COVID-19, S100B was more frequently detected—in other cases, the protein level was low, beyond the sensitivity of the test. A high result (i.e., CS100B ≥ 156 pg/mL) constituted 5% of records. Moreover, in both groups before COVID-19, S100B was rarely detected, and values in the upper range of the standard curve accounted for 7%. In comparison, percentages in other study groups were as follows: 0% for patients with NfT2DM after SARS-CoV-2 infection and 1% for blood donors. This study showed that the frequency of NSE values above the reference value (≥12 ng/mL) in the blood-donor group (16.7%) was comparable to the frequency of NSE in both the T2DM group before COVID-19 (13.8%) and the NfT2DM group before COVID-19 (12.0%). A significant increase in the number of individuals with NSE values above the reference value was observed in the NfT2DM group after COVID-19 (46%), indicating the impact of SARS-CoV-2 on the level of released NSE. A similar relationship was observed for the average NSE concentration in the studied patient groups. The results, indicating a significantly higher frequency of NSE in the NfT2DM group after COVID-19 compared to the T2DM group after COVID-19, suggest that nephropathy promotes NSE release. Elevated NSE levels in kidney- or urinary-tract-disease patient groups have been previously reported in the literature. It has also been noted that NSE levels increased significantly in patients after kidney transplantation compared to NSE levels in patients before organ transplantation [47]. Furthermore, monitoring NSE levels allowed for the observation of a significant decrease in this protein’s level in the blood two years after kidney transplantation [47]. Changes in NSE levels in the microcirculation of individuals with an early aging phenotype of vessels with calcification and/or fibrosis have also been observed. Sharain and colleagues, in their studies comparing NSE levels in a control group with patient groups (chronic kidney disease and end-stage renal failure), showed a linear increase in NSE concentration with the progression of nephrological disorders, indicating the following trends: 3.97 for patients with chronic kidney disease and 7.46 for patients with end-stage renal failure, with a constant value of 1.00 for the control group. Although the dependence of NSE levels on the progression of nephrological disorders has been confirmed by many studies, the mechanisms causing increased NSE levels due to disease progression are not yet known [48]. Additionally, NSE levels in blood serum may vary in different metabolic diseases, making this protein potentially useful as a prognostic marker. NSE is considered a marker for one of the complications of diabetes, i.e., diabetic neuropathy. Low levels of NSE mRNA were found primarily in the group with NfT2DM compared to the T2DM group and healthy individuals [49]. This observation may be significant in interpreting the results presented in this work—the significant decrease in NSE in the T2DM group after COVID-19 compared to the T2DM group before COVID-19 may result from the progression of diabetic neuropathy. Studies by another team, although not confirming this dependence [50].

Similar observations were noted for NE levels in the different groups of patients studied in this work. Since NE was undetectable in many blood serum samples, a “threshold” concentration of 385 pg/mL was conventionally adopted. This value corresponds to slightly less than half of the value determined by the MyAssays program, which set a cutoff point at 780 pg/mL, dividing the dataset into two groups (those with the lowest and highest results). The cutoff concentration was set to half this value due to the nature of NE as an indicator enzyme, which is typically either absent or present at low levels in the blood. Similar to the findings for NSE, it was found that the frequency of elevated NE (i.e., NE > 385 pg/mL) in groups such as voluntary blood donors, patients with T2DM before COVID-19, and patients with advanced-stage nephropathy before COVID-19 was comparable and lower than in the group of patients with advanced nephropathy who had recovered from COVID-19. This difference was statistically significant. Additionally, it was observed that the number of patients with NE > 385 pg/mL increased in the group of T2DM patients who had recovered from COVID-19, compared to those before the pandemic; however, this increase was not statistically significant.

The results presented in this study were consistent with observations in the literature. It is known that excessive activation of immune cells leads to an increase in NE levels, which is associated with severe inflammation and multi-organ damage [51]. The COVID-19 pandemic has demonstrated a connection between NE levels and lung condition, as well as the involvement of this enzyme in severe lung inflammation, such as that caused by SARS-CoV-2 [4]. Additionally, previous studies examining the relationship between NE and kidney dysfunctions have shown that NE levels increase as nephropathy progresses. In studies conducted on rats, it was shown that lithotripsy waves increased NE levels, leading to the initiation and progression of inflammation, tubular dilation, hemorrhages, and damage to glomeruli and renal interstitial tissue. The use of an NE inhibitor—sivelestat—reduced these injuries [52]. Thus, the results of this study suggest that elevated NE levels may be associated with worsening kidney conditions and that NE may serve as a marker of kidney and/or urinary tract damage. Similar conclusions were drawn in another study in which NE inhibitors like sivelestat were used in septic rats, showing a regression in markers of kidney damage and an improvement in kidney function [53]. These studies, demonstrating NE inhibition through sivelestat, show a reduction in inflammation and tissue damage, highlighting the key role of neutrophil elastase in kidney damage. The present study also confirmed that NE > 385 pg/mL was more common in patients with advanced nephropathy (both before and after COVID-19) compared to patients with T2DM or voluntary blood donors. Metabolic and inflammatory pathways are closely interconnected. Type 2 diabetes is characterized by chronic inflammation, with high levels of inflammatory mediators functioning in a feedback loop [54]. Some studies in the literature suggest that hyperglycemia significantly lowers NE levels and enhances coagulation [55]. For instance, it has been shown that in healthy rats, inducing systemic inflammation by intravenous injection of bacterial endotoxins from Escherichia coli increased cytokine release, neutrophil activation, and coagulation factors, while hyperglycemia reduced neutrophil elastase secretion. This study also showed that NE > 385 pg/mL was much less frequent in the group of T2DM patients. Moreover, the significantly higher NE concentration may be due to vascular complications, often resulting from more advanced T2DM. It was expected that the highest percentage of NE > 385 pg/mL would be found in patients with advanced nephropathy who had recovered from COVID-19. The conducted studies confirmed this hypothesis. Since immunosenescence, or the weakening of the immune profile, occurs during aging, some individuals may experience an imbalance between pro- and anti-inflammatory factors, resulting in chronic inflammation. Neutrophil dysfunctions in aging include ineffective bacterial infection elimination, impaired chemotaxis, diminished phagocytosis, reduced ROS production, and increased degranulation, leading to the release of enzymes such as neutrophil elastase. Interestingly, despite NE being essential for neutrophil extracellular traps (NETs), the formation of these structures is defective in older individuals, reducing the ability to capture pathogens [56]. Since the average age within each study group was similar and the focus was on an overall concentration analysis within each group, age should not have influenced NE levels. However, for future studies, it would be advisable to expand the study group and perform subgroup analyses based on age.

## 4. Materials and Methods

A total of 319 serum samples were analyzed, obtained from patients of the Jan Mikulicz-Radecki University Clinical Hospital in Wrocław in Poland and individuals who donated blood at the Regional Blood Donation and Blood Therapy Center of Professor Tadeusz Dorobisz in Wrocław in Poland. Blood was collected from individuals aged from 45 to 89 years, with a gender ratio of 1:1. The patients were divided into 5 groups: (1) patients with type 2 diabetes treated at the Clinic of Angiology, Diabetology, and Hypertension of University Clinical Hospital in Wroclaw, Poland, from whom serum and plasma were collected before COVID-19, i.e., in 2019/2020 (n = 59 samples); (2) patients with type 2 diabetes treated at the Clinic of Angiology, Diabetology, and Hypertension of University Clinical Hospital in Wroclaw, Poland, from whom serum and plasma were collected after recovering from COVID-19, i.e., in 2022–2024 (n = 62 samples); (3) patients with NfT2DM (most commonly diabetic), i.e., stage IV or V, treated at the Clinic of Nephrology and Transplant Medicine of University Clinical Hospital in Wroclaw, Poland, from whom serum and plasma were collected before COVID-19, i.e., in 2019/2020 (n = 51 samples); (4) patients with NfT2DM (most commonly diabetic), i.e., stage IV or V, treated at the Clinic of Nephrology and Transplant Medicine of University Clinical Hospital in Wroclaw, Poland, from whom serum and plasma were collected after recovering from COVID-19, i.e., in 2022–2024 (n = 55 samples); and (5) control group—voluntary blood donors of the Regional Blood Donation and Blood Therapy Center of Professor Tadeusz Dorobisz in Wrocław in Poland, from whom samples (blood and plasma) were collected before the COVID-19 pandemic, i.e., in 2019, with an average age of 51 years (n = 98 samples). The patients did not suffer from any other disease that could have a significant impact on the obtained results. People with psychiatric disorders and cancer were excluded from the study. The material obtained from patients was processed within the shortest possible time, i.e., within one hour of collection, appropriately secured, anonymized, and stored at −80 °C. The acquisition of biological material was approved by the Bioethics Committee of the Medical University of Wrocław of Poland (approval numbers KB 187/2019 and KB 780/2022). Blood was collected over a 3-year interval (the pre-COVID-19 group was collected in 2019/2020, and the post-COVID-19 group in 2022–2024).

The concentrations of NE, S100B, and NSE were determined using the methodologies developed by the manufacturers of the ELISA tests (respectively: Human Neutrophil Elastase ELISA Kit, cat. no. E0778Hu, SUNLONG BIOTECH CO (Hangzhou, China); Human Protein S100-B ELISA Kit, cat. no. E2200HU, SUNLONG BIOTECH CO; and Human NSE (Neuron Specific Enolase) ELISA Kit, cat. no. DKO 073, DiaMetra (Spello, Italy). The tests used were characterized by high sensitivity and a wide range of detectable concentrations: for NE, sensitivity was >31 pg/mL, with a range from 78 to 5000 pg/mL; for S100B, sensitivity was >7.8 pg/mL, with a concentration range from 15.6 to 1000 pg/mL; and for NSE, sensitivity was >0.19 ng/mL, with a concentration range from 0.25 to 25 ng/mL. Absorbance values were read using a BioTek Synergy H1 Multimode Reader (Agilent, Santa Clara, CA, USA).

Differences between the two sexes in age and concentrations of NE, NSE, S100B, and creatinine were assessed with the Mann–Whitney U test, while the monotonic correlations between NE; NSE; and age, total protein, creatinine, and urea were analyzed with use of Spearman correlation coefficient.

## 5. Conclusions

The presented study shows that NE, NSE, and S100B are potential components of a prognostic model for long-term complications caused by COVID-19 in patients with comorbid conditions such as type 2 diabetes (T2DM) or advanced diabetic nephropathy (NfT2DM). This is indicated by the significantly higher occurrence of elevated levels of these proteins in patients with T2DM or NfT2DM who have recovered from COVID-19, compared to patients with the same comorbidities whose blood was collected before COVID-19. Elevated levels of S100B were found more frequently in patients with T2DM after COVID-19, while NE and NSE were elevated in those with NfT2DM. The study also proposed threshold values for NE and S100B, namely 385 pg/mL and 7.8 pg/mL, respectively. The research is preliminary, aimed at assessing the feasibility of studying NE, NSE, and S100B as elements of a prognostic model for long-term complications caused by COVID-19. The next stage of the research will involve a statistical analysis. This analysis will allow for the examination of potential quantitative and qualitative relationships between NE, NSE, and S100B levels and standard indicators of inflammation, metabolic disorders, and clinical parameters. These studies will also take into account the diabetes treatment being used. Parameters showing significant differences will undergo multivariate analyses in order to develop prognostic models for more severe cases of post-COVID-19.

## Figures and Tables

**Figure 1 ijms-25-11791-f001:**
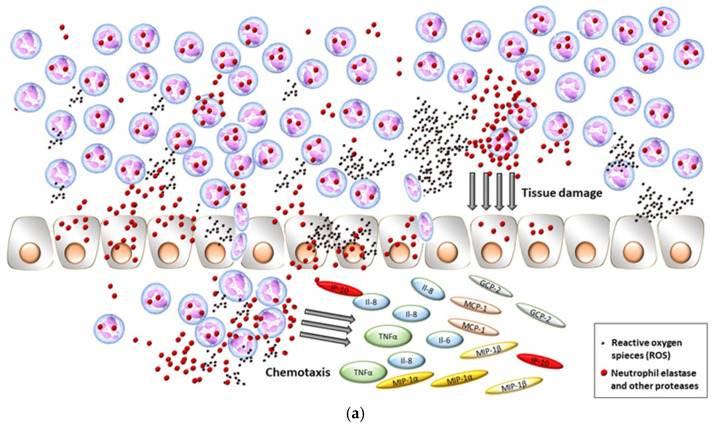
Molecular pathways of the action of (**a**) NE, (**b**) S100B, and (**c**) NSE.

**Figure 2 ijms-25-11791-f002:**
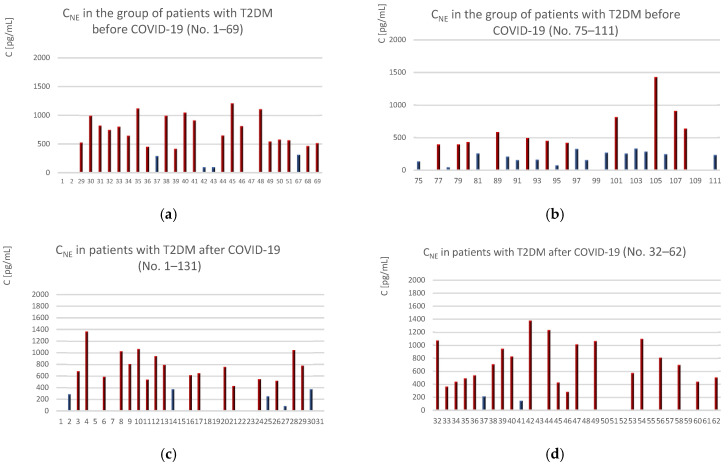
Values of NE concentrations in blood serum: (**a**,**b**) patients with T2DM from whom blood was collected before COVID-19 and (**c**,**d**) patients with T2DM who recovered from COVID-19. Samples with NE levels exceeding the established threshold of 385 pg/mL are marked in red, while samples with NE levels below 385 pg/mL are marked in blue.

**Figure 3 ijms-25-11791-f003:**
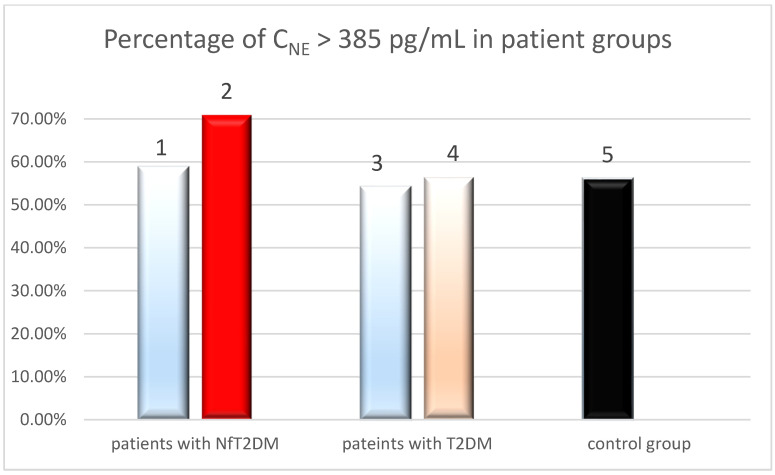
The percentage of patients with NE levels > 385 pg/mL in the following groups: (1) with advanced stage nephropathy (NfT2DM) before COVID-19, (2) with advanced stage nephropathy (NfT2DM) after COVID-19, (3) with T2DM before COVID-19, (4) with T2DM after COVID-19; and (5) of control group.

**Figure 4 ijms-25-11791-f004:**
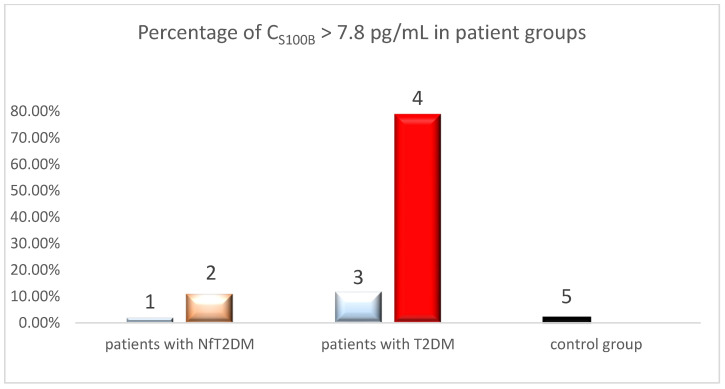
The percentage of patients with S100B levels ≥ 7.8 pg/mL in the following groups: (1) with advanced stage nephropathy (NfT2DM) before COVID-19, (2) with advanced stage nephropathy (NfT2DM) after COVID-19, (3) with T2DM before COVID-19, (4) with T2DM after COVID-19, and (5) of control group.

**Figure 5 ijms-25-11791-f005:**
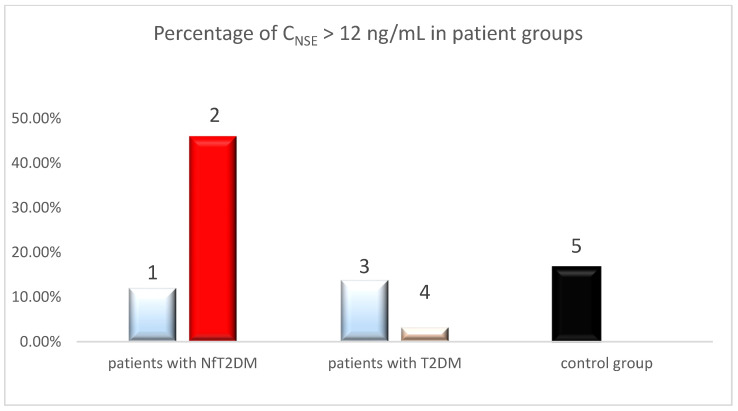
Percentage of patients with NSE ≥ 12 ng/mL in the groups of patients: (1) with advanced stage nephropathy (NfT2DM) before COVID-19, (2) with advanced stage nephropathy (NfT2DM) after COVID-19, (3) with T2DM before COVID-19, (4) with T2DM after COVID-19, and (5) of control group.

**Figure 6 ijms-25-11791-f006:**
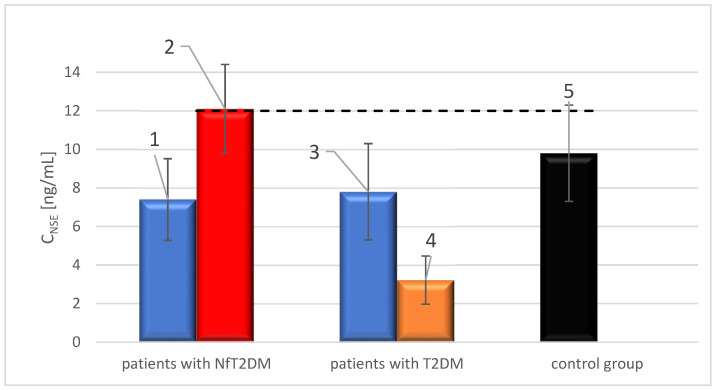
Average NSE concentration in the groups of patients: (1) with advanced stage nephropathy (NfT2DM) before COVID-19, (2) with advanced stage nephropathy (NfT2DM) after COVID-19, (3) with T2DM before COVID-19, (4) with T2DM after COVID-19, and (5) of control group. The black dashed line indicates the upper limit of the reference range for NSE, as specified by the diagnostic test manufacturer.

**Figure 7 ijms-25-11791-f007:**
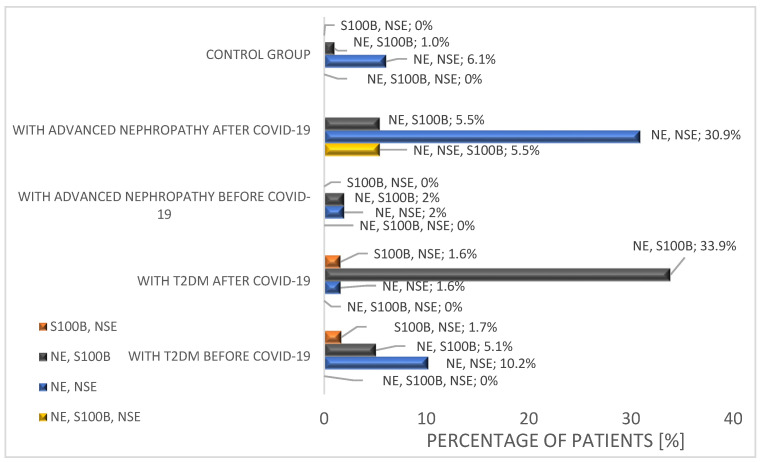
Percentage of patients in a given group who had simultaneous increases in at least 2 of the 3 assayed proteins, i.e., NE, NSE, and S100B.

**Table 1 ijms-25-11791-t001:** Values of NE, NSE, and S100B in the studied groups, stratified by sex.

Variable	Group	Female						Male						*p*
		n	Median Value	Min	Max	1st Q	3rd Q	n	Median Value	Min	Max	1st Q	3rd Q	
Age (years)	A	17	66.00	44.00	85.00	61.00	76.00	39	65.00	31.00	87.00	58.00	71.00	0.548
Creatinine (mg/dL)	A	16	1.15	0.59	2.27	0.82	1.35	37	1.05	0.68	6.44	0.93	1.31	0.901
NE (pg/mL)	A	17	334.50	0.00	1431.00	235.00	800.50	42	427.00	0.00	1210.00	155.10	638.70	0.993
NSE (ng/mL)	A	16	5.94	0.76	14.09	4.29	10.26	42	6.83	2.01	27.22	4.63	9.19	0.698
S100B (pg/mL)	A	17	0.00	0.00	7086.00	0.00	0.00	42	0.00	0.00	18,240.00	0.00	0.00	0.993
Age (years)	B	30	67.50	44.00	81.00	60.00	71.00	29	66.50	39.00	78.00	61.00	71.50	0.868
Creatinine (mg/dL)	B	30	0.76	0.57	1.24	0.69	0.82	29	0.91	0.55	2.82	0.81	1.09	0.006
NE (pg/mL)	B	29	432.00	0.00	1381.00	0.00	650.30	22	578.10	0.00	1074.00	364.60	827.80	0.188
NSE (ng/mL)	B	29	4.68	1.91	13.07	3.35	6.67	22	4.93	2.05	9.58	4.48	6.89	0.527
S100B (pg/mL)	B	29	13.43	5.26	245.40	9.07	20.20	22	13.32	0.00	669.50	7.74	21.29	0.741
Age (years)	C	25	73.00	29.00	92.00	67.00	82.00	26	67.00	49.00	83.00	63.00	71.00	0.038
Creatinine (mg/dL)	C	25	4.38	1.52	8.95	2.84	5.63	26	5.47	2.22	10.18	3.32	7.24	0.072
NE (pg/mL)	C	25	453.80	0.00	1131.00	36.12	895.30	26	443.70	0.00	4390.00	267.40	861.60	0.449
NSE (ng/mL)	C	24	6.12	1.71	20.03	4.32	9.47	26	6.54	2.36	19.62	4.85	9.43	0.870
S100B (pg/mL)	C	25	0.00	0.00	0.00	0.00	0.00	26	0.00	0.00	751.40	0.00	0.00	0.815
Age (years)	D	22	73.00	21.00	90.00	55.00	82.00	33	60.00	29.00	88.00	47.00	70.00	0.013
Creatinine (mg/dL)	D	20	4.31	2.42	10.27	3.75	6.29	32	5.60	1.46	11.27	4.03	8.93	0.126
NE (pg/mL)	D	21	421.40	0.00	851.20	352.40	575.50	34	461.60	0.00	1599.00	372.90	549.50	0.764
NSE (ng/mL)	D	21	11.78	1.70	20.31	7.86	14.29	33	11.85	5.64	23.63	8.94	15.29	0.515
S100B (pg/mL)	D	21	0.00	0.00	87.52	0.00	0.00	34	0.00	0.00	62.04	0.00	0.00	0.576
Age (years)	E	28	49.50	45.00	71.00	46.50	51.50	72	50.00	45.00	65.00	47.00	53.50	0.633
Creatinine (mg/dL)	E	28	0.99	0.59	1.23	0.77	1.06	72	1.06	0.68	1.64	0.94	1.19	0.002
NE (pg/mL)	E	27	466.20	0.00	1251.00	222.80	569.70	62	442.70	0.00	2045.00	211.50	686.70	0.700
NSE (ng/mL)	E	25	9.13	4.59	38.38	7.23	10.86	64	7.96	5.02	33.87	6.70	11.36	0.482
S100B (pg/mL)	E	25	0.00	0.00	11,270.00	0.00	0.00	66	0.00	0.00	0.00	0.00	0.00	0.563

The results are shown for the following groups: patients with T2DM before (A) or after (B) COVID-19, patients with diabetic nephropathy before (C) or after (D) COVID-19, and the control group (E). Q in the columns stands for quartiles.

**Table 2 ijms-25-11791-t002:** Spearman correlation (*p* < 0.05).

Significant Spearman Correlation (*p* < 0.05)	Group	ρ
Creatinine vs. NE	A	−0.296
Creatinine vs. NSE	A	−0.274
Creatinine vs. NE	C	0.314

The results are shown for the following groups: patients with T2DM before COVID-19 (A) and patients with diabetic nephropathy before COVID-19 (C).

## Data Availability

The raw data supporting the conclusions of this article will be made available by the authors upon request.

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
