# Peer review of "Neutrophil Elastase, Neuron-Specific Enolase, and S100B Protein as Potential Markers of Long-Term Complications Caused by COVID-19 in Patients with Type 2 Diabetes Mellitus (T2DM) and Advanced Stage of Diabetic Nephropathy (NfT2DM)—Observational Studies"

_ijms, 2024, doi:10.3390/ijms252111791_

Round 1
Reviewer 1 Report
Comments and Suggestions for Authors
The investigators of the present study tried to demonstrate that NE, NSE, and S100B could be potential predictive biomarkers for long COVID-19 in patients with type 2 diabetes (T2DM) or advanced diabetic nephropathy 505 (NfT2DM). The study is interesting and important with significant clinical relevance.
However, a few concerns need to be addressed prior to publication, including:
1. No clinical data were given for the patients.
2. No diagnostic criteria were given for patients with diabetes.
3. Changes in these biomarkers were already present in some subjects. It's known that COVID-19 could change the course or severity of T2DM. Not sure if the changes in the biomarkers were related to the severity of T2DM and /or glucose control.
4. Any relation with age and/or sex?
5. Any relation to kidney function?
Author Response
Summary: The investigators of the present study tried to demonstrate that NE, NSE, and S100B could be potential predictive biomarkers for long COVID-19 in patients with type 2 diabetes (T2DM) or advanced diabetic nephropathy 505 (NfT2DM). The study is interesting and important with significant clinical relevance. However, a few concerns need to be addressed prior to publication, including:
Thank you for reviewing our manuscript, for your time, and for your very valuable comments. They have helped make the manuscript clearer and more comprehensible. We are extremely motivated by the fact that, in the opinion of the Reviewer, our manuscript is interesting, important, and its findings have significant clinical relevance. We have addressed all of the Reviewer’s comments, and we hope that the changes we have made meet the Reviewer’s expectations. If not, we are more than willing to make further adjustments. We also share the goal of making our article as valuable as possible for every reader.
Comments 1. No clinical data were given for the patients.
Response 1. Thank you very much for this comment. Although the study is not a statistical one, but rather an observational one, we have indeed added data and presented it in Tables 1-5 to make it more credible, as we intend to provide a statistical analysis in our next work. At this stage, we deliberately did not conduct a statistical analysis, as we plan to include a third follow-up – collecting biological material from the same patients as in the second follow-up (with mortality expected to be significantly lower) – as well as additional predictive markers that may have diagnostic relevance, such as IL-17, proteinase 3, methylglyoxal, various types of advanced glycation end products, and sRAGE. At that time, we will also incorporate clinical parameters (diagnoses) such as microangiopathy, retinopathy, macroangiopathy, autoimmune disorders, and neurological disorders. We plan to publish a series of papers that will include multivariate analyses and, if the results indicate, present predictive models with specific relationships between the studied parameters. The purpose of this publication is to present our observations concerning NE, NSE, and S100B, in the hope that they might guide other research teams in their studies on COVID-19 and POST-COVID-19 complications.
Comments 2. No diagnostic criteria were given for patients with diabetes.
Response 2. Thank you for this valuable comment. At the Reviewer's suggestion, a short statement: The patients did not suffer from any other disease that could have a significant impact on the obtained results. People with psychiatric disorders and cancer were excluded from the study has been added to the sentence: A total of 319 serum samples were analyzed, obtained from patients of the Jan Mikulicz-Radecki University Clinical Hospital in Wroclaw and individuals who donated blood at the Regional Blood Donation and Blood Therapy Center of Professor Tadeusz Dorobisz in Wroclaw. Blood was collected from individuals aged 45 to 89 years, with a gender ratio of 1:1. The patients were divided into 5 groups: 1) Patients with type 2 diabetes treated at the Clinic of Angiology, Diabetology, and Hypertension, from whom serum and plasma were collected before COVID-19, i.e., in 2019/2020 (n=59 samples); 2) Patients with type 2 diabetes treated at the Clinic of Angiology, Diabetology, and Hypertension, from whom serum and plasma were collected after recovering from COVID-19, i.e., in 2022-2024 (n=62 samples); 3) Patients with advanced stage nephropathy (most commonly diabetic), i.e., stage IV or V, treated at the Clinic of Nephrology and Transplant Medicine, from whom serum and plasma were collected before COVID-19, i.e., in 2019/2020 (n=51 samples); 4) Patients with advanced stage nephropathy (most commonly diabetic), i.e., stage IV or V, treated at the Clinic of Nephrology and Transplant Medicine, from whom serum and plasma were collected after recovering from COVID-19, i.e., in 2022-2024 (n=55 samples); 5) CONTROL GROUP – voluntary blood donors, from whom samples (blood and plasma) were collected before the COVID-19 pandemic, i.e., in 2019, with an average age of 51 years (n=98 samples).
Comments 3. Changes in these biomarkers were already present in some subjects. It's known that COVID-19 could change the course or severity of T2DM. Not sure if the changes in the biomarkers were related to the severity of T2DM and /or glucose control.
Response 3. Thank you very much for this valid comment. Given that approximately two years passed between the two blood collections, and that 60% of the samples from the second collection came from the same patients from whom blood was drawn before COVID-19, the severity of diabetes has not changed significantly, and if it has, it is very likely that this was influenced by having had COVID-19. All patients from whom blood was drawn have had type 2 diabetes for at least 10 years and are of similar age. However, all statistical aspects will be analyzed and presented in subsequent cycles of work.
Comments 4. Any relation with age and/or sex?
Response 4. Thank you for this remark. We deliberately chose not to conduct any statistical analyses to check the relationships of the concentrations of the new potential markers of PCS, even the smallest ones. However, we calculated the power of the test to estimate the minimum effect size which could be detected. Based on the assumption of 95% statistical power, an alpha value of 0.05, and N as in our study (319), the cut-off effect size (w) that could be observed as statistically significant would be 0.2413, using the chi-square test (non-centrality lambda value = 18.5716). This information was gathered from G*Power 3.1.9.7 (University of Düsseldorf). Our intention is to investigate the impact of having had COVID-19 on molecular pathways indicating, for example, multi-organ damage: NE - protease - 3 - IL-17. However, if the Reviewer believes that presenting such simple relationships as the concentrations of NE, NSE, and S100B in relation to age or sex is necessary, we will certainly add this to the manuscript text.
Comments 5. Any relation to kidney function?
Response 5. Thank you for this suggestion. As stated in our response to comment number 4, we would prefer not to analyze the relationships between the concentrations of NE, NSE, and S100B and the values of renal parameters at this stage. We planned to do this in a separate study that would analyze the multifaceted impact of NE, NSE, and S100B concentrations on the course of T2DM and NfT2DM in subsequent follow-ups. Of course, if the Reviewer still believes that it is worthwhile to present these relationships at this stage, we will do so.
Reviewer 2 Report
Comments and Suggestions for Authors
I find the article of some interest, but I have some main concerns about the work.
First, I’m not able to understand if the samples were collected from the same patients in the pre-and post- COVID19 infection in the three groups (T2DM, NfT2DM and control). This is a very important point in my opinion.
Second, even if the authors stated that this research is preliminary, I think that they should calculate the numbers of samples to be processed in order to have a statistical significance.
Third, the authors showed our findings about the variation of plasmatic value of three biomarkers before and after COVID19 infection in diabetic patients and in patients with advanced stage diabetic nephropathy, but it is not so clear the correlation between this demonstration and Long-Term Complications Caused by COVID-19.
Moreover, in the Discussion section, the authors don’t compare their results with others found in the literature and in fact in the text the references from 39 to 55 lack.
Other comments:
Abstract: No comment.
Introduction:
On line 131 I suggest “in patients with type 2 diabetes (T2DM) and advanced stage diabetic nephropathy (NfT2DM)”.
So, in the entire text I suggest using only the abbreviations.
Results:
On lines 153-154 “Samples with NE concentrations exceeding the established threshold value of 385 pg/mL are marked in red.” I think that is better to put this sentence on line 157 because it explains Figure 1.
On lines 181-182 I suggest checking the sentence “These results indicated that the number of patients with NE > 780 pg/mL with T2DM before COVID-19 increased by 5 percentage points 181 (p.p.) compared to the group with T2DM after COVID-19.” Are “before” and “after” used correctly?
On lines 191 there are two “was”, please correct it.
On lines 201-202 I suggest eliminating “The significantly higher NE concentration may be due to vascular complications, often resulting from more advanced type 2 diabetes.” because it is a comment that should be put in the Discussion section.
On lines 206-207 “There was an 8% decrease in concentrations compared to the 206 group with advanced nephropathy before COVID-19.” It’s not clear what 8% was and what the authors mean with “decrease in concentrations”?
Figure 2, Figure 3 Figure 4 and Figure 5: in the legenda the authors report “1)”, “2)”, “3)” and “4)”, but they didn’t match in the figure. Moreover, I suggest modifying “advanced stage nephropathy before COVID-19” with “advanced stage nephropathy before COVID-19 (NfT2DM)”.
I suggest eliminating the lines 262-271: it seems a summary of what the authors reported in the previous sentences.
In Figure 3 I think that is better to eliminate “CONTROL GROUP” above the third bar of the histogram.
Discussion:
I suggest checking these sentences on lines 381-384: “These results suggest that having an infection with a virus like SARS-CoV-2 may cause destabilization of the cytoplasmic membranes of brain cells. Although the highest S100B concentration in the group of T2DM 383 patients whose blood was collected after COVID-19 was only 700 pg/mL, this protein…”.
I suggest eliminating lines 434-443 because in my opinion these sentences didn’t add anything to the text.
Materials and Methods:
Why authors didn’t report also for NSE the values of its sensitivity and of the concentration range?
Author Response
Summary: I find the article of some interest, but I have some main concerns about the work.
We would like to sincerely thank the Reviewer for thoroughly reviewing the manuscript and providing valuable and thoughtful comments and suggestions. We are pleased that the Reviewer had a positive reception of our article, which motivates us to continue our work. We have addressed all of them, and in our opinion, they have greatly improved the quality of the manuscript, making it more clear and substantial. We hope that our responses meet both the Reviewer’s and the Editor’s expectations. We remain open to any further suggestions and feedback.
Comments 1: First, I’m not able to understand if the samples were collected from the same patients in the pre-and post- COVID19 infection in the three groups (T2DM, NfT2DM and control). This is a very important point in my opinion.
Response 1: Thank You for this suggestion. We fully agree with the Reviewer that the origin of samples from the same patients (two follow-ups) is an important issue. Our original intention (immediately after the pandemic) was indeed to collect blood samples primarily from the same patients from whom we had collected samples just before the COVID-19 pandemic. In the presented study, about 60% of the samples from patients who had recovered from COVID-19 came from the same individuals (though the groups are not equal). We began collecting blood samples before the pandemic in mid-2019, and we managed to collect only 110 samples before the pandemic started, which forced us to stop the collection for a while.
Unfortunately, the pandemic led to over 60% of patients with advanced diabetic nephropathy (from whom we had collected material before COVID-19) not surviving the disease, so it was impossible to obtain biological material from them post-pandemic. We therefore began collecting material from other individuals with advanced diabetic nephropathy, with the aim of collecting samples again from them in another 2.5-year interval (2022-2024). Due to the interesting (and even surprising to us) results obtained at this stage, which indicate the potential diagnostic significance of NE, S100B, and NSE, we decided that it was worth publishing our observations now. These may be helpful to other researchers looking for potential molecular markers of COVID-19 and POST-COVID-19 (PCS). Given the growing number of patients worldwide suffering from PCS, such observations may save other researchers time in identifying potential markers of COVID-19 and PCS.
At this stage, we deliberately did not conduct a statistical analysis, as we plan to include a third follow-up – collecting biological material from the same patients as in the second follow-up (with mortality expected to be significantly lower) – as well as additional predictive markers that may have diagnostic relevance, such as IL-17, proteinase 3, methylglyoxal, various types of advanced glycation end products, and sRAGE. At that time, we will also incorporate clinical parameters (diagnoses) such as microangiopathy, retinopathy, macroangiopathy, autoimmune disorders, and neurological disorders. We plan to publish a series of papers that will include multivariate analyses and, if the results indicate, present predictive models with specific relationships between the studied parameters.
The purpose of this publication is to present our observations concerning NE, NSE, and S100B, in the hope that they might guide other research teams in their studies on COVID-19 and POST-COVID-19 complications.
Considering that the valid question posed by the Reviewer may be raised by other readers as well, we have added the following sentence to the "Materials and Methods" section: Due to the high mortality rate among patients with NfT2DM caused by COVID-19, we were able to obtain material from only about 60% of the same patients for the second part of the study (in the years 2022-2024), i.e., right after recovering from COVID-19. The rest of the samples came from patients who had also recovered from COVID-19, but from whom we did not have pre-COVID19 biological material.
Comments 2: Second, even if the authors stated that this research is preliminary, I think that they should calculate the numbers of samples to be processed in order to have a statistical significance.
Response 2: We are content to see a response with a request for providing the information from power analysis as we believe that this remark may originate from sheer spark of interest in performing research similar to ours. Based on an assumption of 95% statistical power, alpha value of 0.05, and N as in our study (319), the cut-off effect size (w) which could be observed as statistically-significant would be 0.2413, given the use of a chi-square test (non-centrality lambda value = 18.5716). This information was gathered from G*Power 3.1.9.7 (Universität Düsseldorf).
Comments 3: Third, the authors showed our findings about the variation of plasmatic value of three biomarkers before and after COVID19 infection in diabetic patients and in patients with advanced stage diabetic nephropathy, but it is not so clear the correlation between this demonstration and Long-Term Complications Caused by COVID-19.
Response 3: Thank You for this suggestion. As we have already mentioned in our response to the first comment, the presented study is observational in nature. Diagnostic parameters and assessments indicating the progression of nephropathy will, of course, be included in our next study. To make this assurance more credible, we are attaching Tables: 1 – 5 to the presented work, which shows the characteristics of the patients in each group (we have a more extensive database that also includes therapies used, detailed imaging diagnostic data, etc.). The characteristics in Tables: 1 – 5 include both laboratory tests and clinical parameters.
Comments 4: Moreover, in the Discussion section, the authors don’t compare their results with others found in the literature and in fact in the text the references from 39 to 55 lack.
Response 4: We sincerely thank You for pointing out this error. We are deeply embarrassed and apologize for such a significant mistake. During the process of pasting the discussion onto the template, we accidentally omitted a substantial portion of the text. In the revised version, this section has been completed, and the bibliography and literature references have been thoroughly checked.
Comments 5: On line 131 I suggest “in patients with type 2 diabetes (T2DM) and advanced stage diabetic nephropathy (NfT2DM)”.
So, in the entire text I suggest using only the abbreviations.
Response 5: Thank You for this valuable suggestion. In the current version of the manuscript, abbreviations NfT2DM or T2DM have been applied wherever possible throughout the text
Comments 6: On lines 153-154 “Samples with NE concentrations exceeding the established threshold value of 385 pg/mL are marked in red.” I think that is better to put this sentence on line 157 because it explains Figure 1.
Response 6: Thank You for this remark. Indeed, the sentence pointed out by the Reviewer should be placed in the figure description rather than in the main text, as it explains the figure. The sentence “Samples with NE concentrations exceeding the established threshold value of 385 pg/mL are marked in red” has been moved to the figure caption.
Comments 7: On lines 181-182 I suggest checking the sentence “These results indicated that the number of patients with NE > 780 pg/mL with T2DM before COVID-19 increased by 5 percentage points 181 (p.p.) compared to the group with T2DM after COVID-19.” Are “before” and “after” used correctly?
Response 7: Thank You to the Reviewer for their attentiveness and for pointing out this significant error. The sentence has been revised as follows: The results showed that the number of patients with NE > 780 pg/mL and type 2 diabetes before COVID-19 was 5 percentage points (p.p.) lower compared to the group of T2DM patients after COVID-19.
Comments 8: On lines 191 there are two “was”, please correct it.
Response 8: Thank You for this remark, and we deeply regret making such an editorial error. The sentence has been revised as follows: The highest NE value in the group of patients with NfT2DM before COVID-19 was 1381 pg/mL.
Comments 9: On lines 201-202 I suggest eliminating “The significantly higher NE concentration may be due to vascular complications, often resulting from more advanced type 2 diabetes.” because it is a comment that should be put in the Discussion section.
Response 9: Thank You for this remark. In accordance with the Reviewer's valid suggestion, the sentence: ”The significantly higher NE concentration may be due to vascular complications, often resulting from more advanced type 2 diabetes” has been moved from the "Results" section to the "Discussion" section.
Comments 10: On lines 206-207 “There was an 8% decrease in concentrations compared to the 206 group with advanced nephropathy before COVID-19.” It’s not clear what 8% was and what the authors mean with “decrease in concentrations”?
Response 10: Thank You for bringing this sentence to our attention. Due to its complexity and lack of clarity, as well as the fact that it does not provide significant information to the manuscript, the sentence has been removed.
Comments 11: Figure 2, Figure 3 Figure 4 and Figure 5: in the legenda the authors report “1)”, “2)”, “3)” and “4)”, but they didn’t match in the figure. Moreover, I suggest modifying “advanced stage nephropathy before COVID-19” with “advanced stage nephropathy before COVID-19 (NfT2DM)”.
Response 11: Thank You for pointing out the lack of clarity in labeling the bars on the graph. In the current version, we have followed the Reviewer's valid suggestion. We modified the data labels, replacing the descriptions with numbers 1-5. Additionally, the figure captions have been updated according to the Reviewer's recommendations.
Comments 12: I suggest eliminating the lines 262-271: it seems a summary of what the authors reported in the previous sentences.
Response 12: Thank You for this remark. In accordance with the Reviewer's suggestion, the phrase has been removed.
Comments 13: In Figure 3 I think that is better to eliminate “CONTROL GROUP” above the third bar of the histogram.
Response 13: Thank You for this valid suggestion. The labeling of the control group on the graph was a writing error. It has been removed in the current version.
Comments 14: I suggest checking these sentences on lines 381-384: “These results suggest that having an infection with a virus like SARS-CoV-2 may cause destabilization of the cytoplasmic membranes of brain cells. Although the highest S100B concentration in the group of T2DM 383 patients whose blood was collected after COVID-19 was only 700 pg/mL, this protein…”.
Response 14: Thank You for bringing this to our attention. Since the sentence "These results suggest that having an infection with a virus like SARS-CoV-2 may cause destabilization of the cytoplasmic membranes of brain cells" may be too speculative, it has been removed from the text.
Comments 15: I suggest eliminating lines 434-443 because in my opinion these sentences didn’t add anything to the text.
Response 15: Thank You for the remark. The phrase suggested by the Reviewer: “The average NSE concentration in the blood donor group (9.8 ng/mL) was comparable to the average NSE concentration in both the T2DM group before COVID-19 (7.8 ng/mL) and the NfT2DM group before COVID-19 (7.4 ng/mL), and these values in all of the above groups were within the reference range. Similarly, the average NSE concentration in the T2DM group after COVID-19 (3.22 ng/mL) was within the reference range. In the group of patients with NfT2DM who recovered from COVID-19, the average NSE concentration was 12.1 ng/mL, above the reference value for NSE, which also indicates the impact of SARS-CoV-2 on the level of released NSE” has been removed.
Comments 16: Why authors didn’t report also for NSE the values of its sensitivity and of the concentration range?
Response 16: Thank You for this valid remark. The sensitivity value of the test for NSE (0.19 ng) and the concentration range (0.25 – 25 ng/mL) have been added.
Reviewer 3 Report
Comments and Suggestions for Authors
The manuscript titled “Neutrophil Elastase, Neuron-Specific Enolase, and S100B Protein as Potential Markers of Long-Term Complications Caused by COVID-19 in Patients with Type 2 Diabetes Mellitus (T2DM) and Advanced Stage of Diabetic Nephropathy (NfT2DM)-Observational Studies” by Rabczyńki, M.; et al. is a scientific work where the authors analyzed the serum plasma of more than 300 patients to quantify the levels of neutrophil elastase, S100B protein and neuron-specific enolase. Five different sample populations were taken into account depending the stage of Type 2 Diabetes Mellitus attempting to find the crosstalk with complications caused by COVID-19. This research is interesting, the manuscript is generally well-written and this is a topic of growing interest.
However, it exists some points that need to be addressed (please, see them below detailed point-by-point) to improve the scientific quality of the submitted manuscript paper before this article will be consider for its publication in the International Journal of Molecular Sciences.
1) The authors should consider to add the term “advanced stage diabetic nephropathy” in the keyword list.
2) “Excessive activity (…) thrombosis and a severe course of COVID-19” (lines 66-69). Could the authors provide quantitative data insights according to the worldwide global burdens of COVID-19? This will significantly aid the potential readers to better understand the significance of this research.
3) “The S100B protein is expressed in various types of cells (…) S100B is observed in various conditions (…) level of SARS-CoV-2 and the clinical severity of the disease” (lines 82-97). Here, even if I agree with these statements provided by the authors it should be also mentioned how S100 family negatively impacts on the brain health [1] trough the formation of neurotoxic fibrils [2].
[1] Filipek, A.; et al. S100A6 and Its Brain Ligands in Neurodegenerative Disorders. Int. J. Mol. Sci. 2020, 21, 3979. https://doi.org/10.3390/ijms21113979
[2] Carapeto, A.P.; et al. Morphological and Biophysical Study of S100A9 Protein Fibrils by Atomic Force Microscopy Imaging and Nanomechanical Analysis. Biomolecules 2024, 14, 1091. https://doi.org/10.3390/biom14091091
4) “NSE is primarily found in neurons and activates various cellular pathways (…) production of reactive oxygen species (…) MCP-1 (lines 98-106). The manuscript will benefit of a schematid representation of the action mechanisms and cellular response according to NSE, S100B and NE.
5) Figure 1 (line 156). It may be convenient to define an upper maximum concentration threshold of 2000 pg/mL to better visualize the differences among the examined individuals.
6) Figure 2 (line 226). The standard deviation values should be added. Then, a statistical analysis should be also conducted in order to discern if the observed differences are stastistically significant. Similar comment for the Fig. 3 (line 272) and Fig. 4 (line 308).
7) “4. Materials and Methods. A total of 319 serum samples (…) from individuals aged 45 to 89 years, with a gender ratio of 1:1. The patients were divided into 5 groups (…) average age of 51 years (n=98 samples)” (lines 468-486). Did the patients suffer any other clinical pathology which could affect to the data interpretation? A brief statement should be provided in this regard.
8) “5. Conclusions” (lines 502-514). This section perfectly remarks the most relevant outcomes found by the authors in this work. The authors should furnish a brief statement to discuss about the future action lines to pursue the topic covered in this research.
Author Response
Summary: The manuscript titled “Neutrophil Elastase, Neuron-Specific Enolase, and S100B Protein as Potential Markers of Long-Term Complications Caused by COVID-19 in Patients with Type 2 Diabetes Mellitus (T2DM) and Advanced Stage of Diabetic Nephropathy (NfT2DM)-Observational Studies” by Rabczyńki, M.; et al. is a scientific work where the authors analyzed the serum plasma of more than 300 patients to quantify the levels of neutrophil elastase, S100B protein and neuron-specific enolase. Five different sample populations were taken into account depending the stage of Type 2 Diabetes Mellitus attempting to find the crosstalk with complications caused by COVID-19. This research is interesting, the manuscript is generally well-written and this is a topic of growing interest. However, it exists some points that need to be addressed (please, see them below detailed point-by-point) to improve the scientific quality of the submitted manuscript paper before this article will be consider for its publication in the International Journal of Molecular Sciences.
Thank you for taking the time to review our manuscript, and especially for your thorough and, in our opinion, very valuable comments. It is incredibly motivating for us that the manuscript was received so well. We are also grateful for your insightful suggestions and guidance, which have helped improve the quality of the manuscript. We hope it is now more reader-friendly. If we have not fully met your expectations, we are open to further suggestions and feedback, as we are equally committed to making the manuscript the best it can be.
Comments 1: The authors should consider to add the term “advanced stage diabetic nephropathy” in the keyword list.
Response 1: Thank you for this valid suggestion. We have included the term in the keywords: advanced diabetic nephropathy.
Comments 2: “Excessive activity (…) thrombosis and a severe course of COVID-19” (lines 66-69). Could the authors provide quantitative data insights according to the worldwide global burdens of COVID-19? This will significantly aid the potential readers to better understand the significance of this research.
Response 2: Thank you for this valid suggestion. We have added the quantitative data insights according to the worldwide global burdens of COVID-19: The global spread of COVID-19 has had a profoundly negative impact on global health and the economy. As of 21 February 2023, there have been a total of 757 264 511 confirmed cases of coronavirus worldwide, with 6 850 594 deaths. It is worth noting that the global mortality rate caused by COVID-19 varies considerably depending on a number of factors, including countries, infrastructure of healthcare, rate of vaccination, and the demographics. It is also important to note that people with underlying health conditions, such as chronic respiratory conditions, diabetes, cardiovascular diseases have been particularly susceptible to severe outcomes [3].
Comments 3: “The S100B protein is expressed in various types of cells (…) S100B is observed in various conditions (…) level of SARS-CoV-2 and the clinical severity of the disease” (lines 82-97). Here, even if I agree with these statements provided by the authors it should be also mentioned how S100 family negatively impacts on the brain health [1] trough the formation of neurotoxic fibrils [2].
[1] Filipek, A.; et al. S100A6 and Its Brain Ligands in Neurodegenerative Disorders. Int. J. Mol. Sci. 2020, 21, 3979. https://doi.org/10.3390/ijms21113979
[2] Carapeto, A.P.; et al. Morphological and Biophysical Study of S100A9 Protein Fibrils by Atomic Force Microscopy Imaging and Nanomechanical Analysis. Biomolecules 2024, 14, 1091. https://doi.org/10.3390/biom14091091
Response 3: Thank you for this remark. The sentence suggested by the Reviewer has been added to the text as follows: However, it should be noted that the S100 family has a negative impact on brain health [24] through the formation of neurotoxic fibrils [25].
[24] Filipek, A.; i in. S100A6 i jego ligandy mózgowe w zaburzeniach neurodegeneracyjnych. Int. J. Mol. Sci. 2020, 21, 3979. https://doi.org/10.3390/ijms21113979
[25] Carapeto, A.P.; i in. Morphological and Biophysical Study of S100A9 Protein Fibrils by Atomic Force Microscopy Imaging and Nanomechanical Analysis. Biomolecules 2024, 14, 1091. https://doi.org/10.3390/biom14091091
Comments 4: “NSE is primarily found in neurons and activates various cellular pathways (…) production of reactive oxygen species (…) MCP-1 (lines 98-106). The manuscript will benefit of a schematid representation of the action mechanisms and cellular response according to NSE, S100B and NE.
Response 4: Thank you for this valid suggestion. In the current version of the manuscript, we have included hand-drawn mechanisms of action and cellular responses: NE (Figure 1a), S100B (Figure 1b), and NSE (Figure 1c).
Comments 5: Figure 1 (line 156). It may be convenient to define an upper maximum concentration threshold of 2000 pg/mL to better visualize the differences among the examined individuals.
Response 5: Thank you for this valuable advice. The following sentence has been added: The value of 2000 pg/mL was chosen as the upper maximum concentration threshold.
Comments 6: Figure 2 (line 226). The standard deviation values should be added. Then, a statistical analysis should be also conducted in order to discern if the observed differences are stastistically significant. Similar comment for the Fig. 3 (line 272) and Fig. 4 (line 308).
Response 6: Thank You for this valuable remark. While we agree that providing measures of dispersion (in this case SD) is an important part of a scientific report, the values presented in the mentioned figures are given in percentage (frequency). Therefore, a binomial distribution is attained, for which SD is not used as a measure of dispersion. Therefore, we are afraid to admit that we are unable to meet the demand presented in this remark.
Comments 7: 4. Materials and Methods. A total of 319 serum samples (…) from individuals aged 45 to 89 years, with a gender ratio of 1:1. The patients were divided into 5 groups (…) average age of 51 years (n=98 samples)” (lines 468-486). Did the patients suffer any other clinical pathology which could affect to the data interpretation? A brief statement should be provided in this regard.
Response 7: Thank you for this valuable comment. At the Reviewer's suggestion, a short statement: The patients did not suffer from any other disease that could have a significant impact on the obtained results. People with psychiatric disorders and cancer were excluded from the study has been added to the sentence: A total of 319 serum samples were analyzed, obtained from patients of the Jan Mikulicz-Radecki University Clinical Hospital in Wroclaw and individuals who donated blood at the Regional Blood Donation and Blood Therapy Center of Professor Tadeusz Dorobisz in Wroclaw. Blood was collected from individuals aged 45 to 89 years, with a gender ratio of 1:1. The patients were divided into 5 groups: 1) Patients with type 2 diabetes treated at the Clinic of Angiology, Diabetology, and Hypertension, from whom serum and plasma were collected before COVID-19, i.e., in 2019/2020 (n=59 samples); 2) Patients with type 2 diabetes treated at the Clinic of Angiology, Diabetology, and Hypertension, from whom serum and plasma were collected after recovering from COVID-19, i.e., in 2022-2024 (n=62 samples); 3) Patients with advanced stage nephropathy (most commonly diabetic), i.e., stage IV or V, treated at the Clinic of Nephrology and Transplant Medicine, from whom serum and plasma were collected before COVID-19, i.e., in 2019/2020 (n=51 samples); 4) Patients with advanced stage nephropathy (most commonly diabetic), i.e., stage IV or V, treated at the Clinic of Nephrology and Transplant Medicine, from whom serum and plasma were collected after recovering from COVID-19, i.e., in 2022-2024 (n=55 samples); 5) CONTROL GROUP – voluntary blood donors, from whom samples (blood and plasma) were collected before the COVID-19 pandemic, i.e., in 2019, with an average age of 51 years (n=98 samples).
Comments 8: 5. Conclusions” (lines 502-514). This section perfectly remarks the most relevant outcomes found by the authors in this work. The authors should furnish a brief statement to discuss about the future action lines to pursue the topic covered in this research.
Response 8: Thank you for this important suggestion. A phrase concerning the future research plan has been added to the conclusions section. The updated conclusions section is as follows:
The presented study shows that NE, NSE, and S100B are potential components of a prognostic model for long-term complications caused by COVID-19 in patients with comorbid conditions such as type 2 diabetes (T2DM) or advanced diabetic nephropathy (NfT2DM). This is indicated by the significantly higher occurrence of elevated levels of these proteins in patients with T2DM or NfT2DM who have recovered from COVID-19, compared to patients with the same comorbidities whose blood was collected before COVID-19. Elevated levels of S100B were found more frequently in patients with T2DM after COVID-19, while NE and NSE were elevated in those with NfT2DM. The study also proposed threshold values for NE and S100B, namely 385 pg/mL and 7.8 pg/mL, respectively. The research is preliminary, aimed at assessing the feasibility of studying NE, NSE, and S100B as elements of a prognostic model for long-term complications caused by COVID-19. The next stage of the research will involve statistical analysis. This analysis will allow for the examination of potential quantitative and qualitative relationships between NE, NSE, and S100B levels and standard indicators of inflammation, metabolic disorders, and clinical parameters. These studies will also take into account the diabetes treatment being used. Parameters showing significant differences will undergo multivariate analyses in order to develop prognostic models for more severe cases of post-COVID-19.
Round 2
Reviewer 1 Report
Comments and Suggestions for Authors
The revised manuscript didn't address the major concerns. If the authors didn't have the data to address the concerns, the investigators need to at least discuss them and acknowledge these concerns as weakness and/or limitations.
The relations to age and sex should be part of the data presentation of this manuscript. Should NOT be "saved" for another manuscript.
Author Response
Response to Reviewer 1 Comments
Summary and Comments 1: The revised manuscript didn't address the major concerns. If the authors didn't have the data to address the concerns, the investigators need to at least discuss them and acknowledge these concerns as weakness and/or limitations. The relations to age and sex should be part of the data presentation of this manuscript. Should NOT be "saved" for another manuscript.
We would like to sincerely thank the Reviewer for reviewing our manuscript again. We also apologize for not addressing one of the major concerns raised by the Reviewer in round 1. Indeed, we had several incomplete entries in our database (related to standard diagnostics, sourced from the hospital database), which we have now completed in the presented work. We apologize for this, and we are grateful that we were given the opportunity to respond to this recent comment in the next round. In the current version of the manuscript, we have filled in the data and conducted a preliminary statistical analysis as rightly suggested by the Reviewer. The results of this analysis are presented in section 2.4 (the added text and two new tables are highlighted in yellow). Of course, we have also supplemented the Materials and Methods section with information on the statistical tests used ("Differences between the two sexes in age and concentrations of NE, NSE, S100B, and creatinine were assessed with the Mann-Whitney U test, while the monotonic correlations between NE, NSE and: age, total protein, creatinine, and urea were analyzed using the Spearman correlation coefficient"). We hope that our response is satisfactory. However, if it is not sufficient and requires further elaboration, please provide us with Your comments—we are committed to improving our work, as we want the quality of the publication to be the best possible so that readers can gain the most benefit for their own research.
The added section 2.4 of the manuscript is presented below:
2.4. Between-sex differences in concentration of NE, NSE, S100B, age and creatinine. Significant correlations analyzed between NE and NSE with age, total protein, urea, creatinine among the studied groups
There were no differences between men and women in regards to NE, NSE, and S100B (Table 1). However, women among the two NfT2DM showed lower age compared to men (p = 0.038 and p = 0.013, respectively). Moreover, women among groups: T2DM after COVID-19, and the control group showed higher creatinine concentration compared to men (p = 0.006 and p = 0.002, respectively).
Table 1. Values of NE, NSE and S100B in the studied groups, stratified by sex
Variable |
Group |
Female |
Male |
p |
||||||||||
n |
Median value |
Min |
Max |
1st Q |
3rd Q |
n |
Median value |
Min |
Max |
1st Q |
3rd Q |
|||
Age [years] |
A |
17 |
66.00 |
44.00 |
85.00 |
61.00 |
76.00 |
39 |
65.00 |
31.00 |
87.00 |
58.00 |
71.00 |
0.548 |
Creatinine [mg/dl] |
A |
16 |
1.15 |
0.59 |
2.27 |
0.82 |
1.35 |
37 |
1.05 |
0.68 |
6.44 |
0.93 |
1.31 |
0.901 |
NE [pg/ml] |
A |
17 |
334.50 |
0.00 |
1431.00 |
235.00 |
800.50 |
42 |
427.00 |
0.00 |
1210.00 |
155.10 |
638.70 |
0.993 |
NSE [ng/ml] |
A |
16 |
5.94 |
0.76 |
14.09 |
4.29 |
10.26 |
42 |
6.83 |
2.01 |
27.22 |
4.63 |
9.19 |
0.698 |
S100B [pg/ml] |
A |
17 |
0.00 |
0.00 |
7086.00 |
0.00 |
0.00 |
42 |
0.00 |
0.00 |
18240.00 |
0.00 |
0.00 |
0.993 |
Age [years] |
B |
30 |
67.50 |
44.00 |
81.00 |
60.00 |
71.00 |
29 |
66.50 |
39.00 |
78.00 |
61.00 |
71.50 |
0.868 |
Creatinine [mg/dl] |
B |
30 |
0.76 |
0.57 |
1.24 |
0.69 |
0.82 |
29 |
0.91 |
0.55 |
2.82 |
0.81 |
1.09 |
0.006 |
NE [pg/ml] |
B |
29 |
432.00 |
0.00 |
1381.00 |
0.00 |
650.30 |
22 |
578.10 |
0.00 |
1074.00 |
364.60 |
827.80 |
0.188 |
NSE [ng/ml] |
B |
29 |
4.68 |
1.91 |
13.07 |
3.35 |
6.67 |
22 |
4.93 |
2.05 |
9.58 |
4.48 |
6.89 |
0.527 |
S100B [pg/ml] |
B |
29 |
13.43 |
5.26 |
245.40 |
9.07 |
20.20 |
22 |
13.32 |
0.00 |
669.50 |
7.74 |
21.29 |
0.741 |
Age [years] |
C |
25 |
73.00 |
29.00 |
92.00 |
67.00 |
82.00 |
26 |
67.00 |
49.00 |
83.00 |
63.00 |
71.00 |
0.038 |
Creatinine [mg/dl] |
C |
25 |
4.38 |
1.52 |
8.95 |
2.84 |
5.63 |
26 |
5.47 |
2.22 |
10.18 |
3.32 |
7.24 |
0.072 |
NE [pg/ml] |
C |
25 |
453.80 |
0.00 |
1131.00 |
36.12 |
895.30 |
26 |
443.70 |
0.00 |
4390.00 |
267.40 |
861.60 |
0.449 |
NSE [ng/ml] |
C |
24 |
6.12 |
1.71 |
20.03 |
4.32 |
9.47 |
26 |
6.54 |
2.36 |
19.62 |
4.85 |
9.43 |
0.870 |
S100B [pg/ml] |
C |
25 |
0.00 |
0.00 |
0.00 |
0.00 |
0.00 |
26 |
0.00 |
0.00 |
751.40 |
0.00 |
0.00 |
0.815 |
Age [years] |
D |
22 |
73.00 |
21.00 |
90.00 |
55.00 |
82.00 |
33 |
60.00 |
29.00 |
88.00 |
47.00 |
70.00 |
0.013 |
Creatinine [mg/dl] |
D |
20 |
4.31 |
2.42 |
10.27 |
3.75 |
6.29 |
32 |
5.60 |
1.46 |
11.27 |
4.03 |
8.93 |
0.126 |
NE [pg/ml] |
D |
21 |
421.40 |
0.00 |
851.20 |
352.40 |
575.50 |
34 |
461.60 |
0.00 |
1599.00 |
372.90 |
549.50 |
0.764 |
NSE [ng/ml] |
D |
21 |
11.78 |
1.70 |
20.31 |
7.86 |
14.29 |
33 |
11.85 |
5.64 |
23.63 |
8.94 |
15.29 |
0.515 |
S100B [pg/ml] |
D |
21 |
0.00 |
0.00 |
87.52 |
0.00 |
0.00 |
34 |
0.00 |
0.00 |
62.04 |
0.00 |
0.00 |
0.576 |
Age [years] |
E |
28 |
49.50 |
45.00 |
71.00 |
46.50 |
51.50 |
72 |
50.00 |
45.00 |
65.00 |
47.00 |
53.50 |
0.633 |
Creatinine [mg/dl] |
E |
28 |
0.99 |
0.59 |
1.23 |
0.77 |
1.06 |
72 |
1.06 |
0.68 |
1.64 |
0.94 |
1.19 |
0.002 |
NE [pg/ml] |
E |
27 |
466.20 |
0.00 |
1251.00 |
222.80 |
569.70 |
62 |
442.70 |
0.00 |
2045.00 |
211.50 |
686.70 |
0.700 |
NSE [ng/ml] |
E |
25 |
9.13 |
4.59 |
38.38 |
7.23 |
10.86 |
64 |
7.96 |
5.02 |
33.87 |
6.70 |
11.36 |
0.482 |
S100B [pg/ml] |
E |
25 |
0.00 |
0.00 |
11270.00 |
0.00 |
0.00 |
66 |
0.00 |
0.00 |
0.00 |
0.00 |
0.00 |
0.563 |
The results are shown for the following groups: patients with T2DM before (A) or after (B) COVID-19, patients with diabetic nephropathy before (C) or after (D) COVID-19, and the control group (E). Q in the columns stands for quartiles.
Significant (p < 0.05) weak, positive Spearman correlation was found in NfT2DM patients, before COVID-19 between creatinine and NE (ρ = 0.314). Moreover, T2DM group (before COVID-19) showed weak negative correlations between creatinine and both NE and NSE (ρ = -0.296 and ρ = -0.274, respectively) as shown in Table 2.
Table 2. Spearman correlation (p < 0.05)
Significant Spearman correlation (p < 0.05) |
Group |
ρ |
Creatinine vs. NE |
A |
-0.296 |
Creatinine vs. NSE |
A |
-0.274 |
Creatinine vs. NE |
C |
0.314 |
|
|
|
The results are shown for the following groups: patients with T2DM before COVID-19 (A), patients with diabetic nephropathy before COVID-19 (C).
Reviewer 2 Report
Comments and Suggestions for Authors
I noted that the authors have adequately answered to all my comments and I think that the work was improved.
Author Response
Summary: I noted that the authors have adequately answered to all my comments and I think that the work was improved.
Thank you very much for reviewing our manuscript once again and for the previous valuable comments, which have significantly improved the quality of the manuscript.
Reviewer 3 Report
Comments and Suggestions for Authors
The authors did a great deal of effort to cover all the suggestions raised by the Reviewers. For this reason, the scientific manuscript quality was greatly improved. Based on the significance and novelty of the gathered results, I warmly endorse this work for further publication in the International Journal of Molecular Sciences.
Author Response
Summary: The authors did a great deal of effort to cover all the suggestions raised by the Reviewers. For this reason, the scientific manuscript quality was greatly improved. Based on the significance and novelty of the gathered results, I warmly endorse this work for further publication in the International Journal of Molecular Sciences.
We would like to thank the Reviewer for reviewing our manuscript once again. We are pleased that our responses met the Reviewer's expectations. We are very grateful for the previous comments, which we found to be highly insightful and instrumental in improving the quality of the manuscript. They also motivated us greatly and provided direction for our continued work.
Round 3
Reviewer 1 Report
Comments and Suggestions for Authors
Although the authors didn't address other critical concerns, no further comments.